# Evaluation of Oral Amoxicillin/Clavulanate for Urinary Tract Infections Caused by Ceftriaxone Non-Susceptible Enterobacterales

**DOI:** 10.3390/pharmacy12020060

**Published:** 2024-04-01

**Authors:** Madison E. Salam, Meghan Jeffres, Kyle C. Molina, Matthew A. Miller, Misha Huang, Douglas N. Fish

**Affiliations:** 1Department of Clinical Pharmacy, University of Colorado Skaggs School of Pharmacy and Pharmaceutical Sciences, Aurora, CO 80045, USA; madison.salam@uchealth.org (M.E.S.); meghan.jeffres@cuanschutz.edu (M.J.); 2Department of Pharmacy, UCHealth University of Colorado Hospital, Aurora, CO 80045, USA; 3Department of Pharmacy, Scripps Health, La Jolla, CA 92121, USA; 4Department of Emergency Medicine, University of Colorado School of Medicine, Aurora, CO 80045, USA; 5Department of Pharmacy, Children’s Hospital Colorado, Aurora, CO 80045, USA; matthew.miller2@childrenscolorado.org; 6Division of Infectious Diseases, Department of Medicine, University of Colorado School of Medicine, Aurora, CO 80045, USA; misha.huang@cuanschutz.edu; 7Department of Medicine–Infectious Diseases, UCHealth University of Colorado Hospital, Aurora, CO 80045, USA

**Keywords:** UTI, ESBL, amoxicillin-clavulanate, ceftriaxone non-susceptible

## Abstract

Urinary tract infections (UTIs) are one of the most common infections and are frequently caused by Gram-negative organisms. The rise of resistant isolates has prompted evaluation of alternative therapies, including amoxicillin-clavulanate which has potent activity against Ambler class A enzymes. This study sought to evaluate clinical outcomes of patients with ceftriaxone non-susceptible UTIs receiving amoxicillin-clavulanate or standard of care (SOC). This was a single-center, retrospective, cohort study of adult patients with urinary tract infections caused by a ceftriaxone non-susceptible pathogen who received amoxicillin-clavulanate or SOC. The primary outcome was clinical failure at 90 days. Secondary outcomes included time to failure, isolation of a resistant organism, and hospital length of stay. Fifty-nine patients met study inclusion: 26 received amoxicillin/clavulanate and 33 received SOC. Amoxicillin-clavulanate recipients did not have higher failure rates compared to SOC recipients. For patients requiring hospital admission, hospital length of stay was numerically shorter with amoxicillin-clavulanate. The frequency of amoxicillin-clavulanate and carbapenem-resistant organisms did not differ significantly between groups. Amoxicillin-clavulanate may be a useful alternative therapy for the treatment of ceftriaxone non-susceptible *Enterobacterales* UTIs.

## 1. Introduction

The burden of infections caused by extended-spectrum beta-lactamase (ESBL)-producing organisms is significant, and prevalence has been identified to be up to 17.2% in urinary tract infections (UTI) alone [1,2]. Patients that are infected with ESBL Enterobacterales have higher mortality rates than their counterparts infected with non-ESBL isolates; therefore, prompt and appropriate treatment is of the utmost importance [3]. There is, however, a compelling need to balance appropriate therapy with antimicrobial stewardship to prevent further development of resistance. Oral treatment options for ESBL-producing organisms are limited due to the hydrolysis of common oral beta-lactams by ESBL enzymes. Non-beta-lactam oral antibiotics such as fluoroquinolones, trimethoprim/sulfamethoxazole, and nitrofurantoin are frequently suboptimal due to their associated adverse events, drug interactions, and lack of ability to concentrate in the renal parenchyma [4]. Additionally, epidemiologic studies show that in urinary bacterial isolates that express ESBL enzymes, co-expression of resistant phenotypes is common with fluoroquinolones (74.1%), sulfamethoxazole-trimethoprim (60.3%), and nitrofurantoin (35.5%) [5]. Thus, carbapenems are frequently the only available option for treatment, but use is complicated by the need for parenteral administration, often requiring hospital admission or outpatient parenteral antimicrobial therapy (OPAT), and desire to prevent further resistance. Therefore, patients may experience otherwise avoidable contact with the healthcare system, and many limited resources are expended on a disease state that could otherwise be managed with outpatient use of oral antimicrobials.

Recent Infectious Diseases Society of America (IDSA) guidance regarding the treatment of ESBL-producing Gram-negative organisms do not recommend amoxicillin/clavulanate for the treatment of ESBL UTIs [6]. Clavulanate may restore the in vitro activity of amoxicillin against resistant organisms owing to potent affinity for Ambler beta-lactamase class A enzymes including common ESBL enzymes: TEM- and SHV-type variants, and CTX-M [7]. The significant urinary elimination of amoxicillin and clavulanate result in urine concentrations that may be sufficient to successfully inhibit EBSL, resulting in a clinical cure. Experience with amoxicillin/clavulanate is limited to single-arm observational evidence [8]. Given the overall paucity of data and dire need for more options for the treatment of ESBL-producing isolates in clinical practice, we sought to compare the clinical outcomes among patients with UTIs caused by ceftriaxone non-susceptible Enterobacterales treated with either amoxicillin/clavulanate or standard of care. 

## 2. Materials and Methods

### 2.1. Patients

We conducted a retrospective cohort of adult patients treated for a urinary tract infection within a single healthcare system in both the inpatient and outpatient settings from 1 January 2012, to 1 August 2022. Patients were included if they had uncomplicated cystitis, complicated cystitis or uncomplicated pyelonephritis with selected Enterobacterales (*Escherichia coli*, *Klebsiella pneumoniae*, *Klebsiella oxytoca*, *Proteus mirabilis*, or *Proteus vulgaris*) that were either intermediate or resistant (e.g., non-susceptible) to ceftriaxone using 2010 CSLI breakpoints (MIC > 2) as tested using the BD Phoenix^TM^ panel and who received at least 72 h of in vitro active antibiotics. Patients who received active lead-in antibiotics for >50% of the treatment course were excluded. Additional exclusion criteria included: definitive use of combination antimicrobials, concurrent bloodstream infection, nephrolithiasis, renal abscesses, nephrostomy tubes, or a previous qualifying UTI episode within 90 days that was already captured in the dataset. Following local institutional review board requirements, patients over ninety years of age, pregnant, incarcerated, or those with physical or cognitive impairment were excluded. Organisms that harbor chromosomal AmpC, including *Enterobacter cloacae*, *Klebsiella aerogenes*, *Hafnia alvei*, and *Citrobacter freundii* were not included due to the inability of ceftriaxone non-susceptibility to differentiate between AmpC and ESBL without additional testing. *Serratia marcescens* and *Citrobacter koseri* were not included due to their intrinsic resistance to ampicillin and low prevalence of being the causative pathogens in UTIs [9]. Patients with molecular detection of a carbapenemase (KPC, IMP, VIM, NDM, and OXA-48) by Xpert^®^ Carba-R (Cepheid, Sunnyvale, CA, USA) were excluded to limit heterogeneity.

### 2.2. Definitions

Uncomplicated cystitis was defined as a symptomatic urinary tract infection that did not meet any criteria for complicated cystitis or pyelonephritis as defined later in this section. Symptoms were required to be documented in the electronic medical record and included dysuria, urinary frequency or urgency, or suprapubic pain. Alternatively, in patients unable to endorse subjective symptoms, they were required to have a systemic sign of infection including leukocytosis, hemodynamic instability, or fever as well as both pyuria (defined as urine white blood cells > 10 cells/mm^3^) and an organism identified from urine culture to be included [10]. Complicated cystitis was defined as a symptomatic urinary tract infection in patients who were either over 65 years old, male, diabetic, immunocompromised, or had urinary tract abnormalities (ileal conduit, suprapubic catheter, indwelling Foley catheter, self-catheterization, or benign prostatic hyperplasia) without signs and symptoms of pyelonephritis [11]. Pyelonephritis was defined as at least one of the following: renal computed tomography (CT) scan or ultrasound suggestive of pyelonephritis, documentation of infection as pyelonephritis in the electronic medical record, or documented presence of costovertebral angle tenderness or flank pain [12]. In vitro active therapy was defined as an antimicrobial reported as susceptible per Clinical and Laboratory Standards Institute (CLSI) guidelines, including amoxicillin/clavulanate at a breakpoint of <16/8 mg/L. Standard of care was defined as any in vitro active therapy used in accordance with the IDSA Gram-negative resistance guidelines [6]. 

Lead-in therapy was defined as therapy received prior to definitive antibiotics. Combination definitive therapy was defined as two or more in-vitro active agents for definitive therapy. Immunocompromised was defined as receipt of chemotherapy within 30 days, daily steroids with a prednisone equivalent of over 20 mg for at least 14 days, receipt of solid organ transplant, receipt of hematopoietic stem cell transplant within 2 years, receipt of other immunosuppressive agents, or HIV infection with a CD4 <200 cells/mm^3^ [13]. 

### 2.3. Outcomes

The primary outcome was clinical failure within 90 days after the end of therapy. Clinical failure was a composite endpoint comprised of either recurrence of symptoms that prompted treatment with antibiotics, a repeat urine culture positive for the same organism as the index infection, or both.

Secondary outcomes included individual components of the primary outcome, time to clinical failure, isolation of an organism from any site that was carbapenem-resistant or resistant to amoxicillin/clavulanate within 90 days after the end of therapy, and hospital length of stay.

### 2.4. Data Management and Statistics

A microbiology report using dates from 1 January 2012 to 1 August 2022 was used to identify urine cultures with qualifying organisms. Data were manually collected via retrospective chart review. Statistical analyses were performed using R v.4.1.2 (R Core Team (2021)). Chi-squared or Fisher’s exact tests were used for categorical variables as appropriate, and Student’s *t*-test was used for continuous variables. In this noninferiority assessment, using a noninferiority limit of 10% and assuming a clinical success rate of 93% in each group [14], 81 patients in each arm would be required to meet 80% power at a significance level of 5%. 

## 3. Results

One-hundred and sixty-six patients were identified on the microbiology report, and 59 met study inclusion (Figure 1). The majority of patients were excluded due to the positive urine culture not representing acute cystitis or pyelonephritis as defined above or due to lead-in therapy lasting more than 72 h.

Of the included patients, the mean age was 60.0 years, and the cohort was predominantly female (66.1%). The majority (64.4%) of patients had at least one positive urine culture in the preceding year. Infection types included complicated cystitis (64%), pyelonephritis (32%), and, less commonly, uncomplicated cystitis (3%). The predominant pathogen identified was *E. coli* (81.4%); the remaining isolates included in this analysis were *K. pneumoniae*, and the distribution did not differ between therapy groups. Of the 59 patients included, 26 (44%) received definitive therapy with amoxicillin/clavulanate. Significantly fewer patients receiving amoxicillin/clavulanate were immunocompromised (11.5% vs. 36.4%, *p* = 0.030). Amoxicillin/clavulanate dosing regimens among those with a calculated creatinine clearance 30 mL/min or greater were 875/125 mg every 12 h (90.5%), 500/125 mg every 12 h (4.8%), and 500/125 mg every 8 h (4.8%). Among those with a creatinine clearance less than 30 mL/min, three patients (60.0%) received 875/125 mg every 12 h, and two patients received 500/125 mg every 24 h (40%). 

The remaining 33 patients received standard of care with agents that included: ertapenem (60.6%), meropenem (15.2%), trimethoprim/sulfamethoxazole (6.1%), levofloxacin (6.1%), amikacin (6.1%), piperacillin/tazobactam (3.0%), or nitrofurantoin (3.0%). 

The majority of patients (66.1%) received lead-in antimicrobials. Lead-in therapy was active significantly more often in the standard-of-care group: 63.2% of the standard-of-care group versus 31.6% of the amoxicillin/clavulanate group (*p* = 0.037). Definitive treatment duration was, on average, 7.0 days in the amoxicillin/clavulanate group and 8.8 days in the standard-of-care group (Table 1).

The composite primary outcome of clinical failure at 90 days occurred in five (19.2%) of amoxicillin/clavulanate recipients and 10 (30.3%) standard-of-care recipients (*p* = 0.332). Reasons for treatment failure (symptom relapse prompting retreatment, repeat culture with same infecting organism, or both) did not differ significantly between groups, and failure was driven mostly by symptom relapse without repeat culture in both groups. No baseline characteristics were associated with treatment failure (*p* > 0.05). The mean time to treatment failure was 31.0 + 23.8 days in the amoxicillin/clavulanate group and 23 + 36.0 days in the standard-of-care group. In amoxicillin/clavulanate recipients, 69.2% of patients were inpatient at the time of culture collection, compared to 54.5% of standard-of-care recipients (*p* = 0.251). Among patients who were hospitalized at the time of positive urine culture, hospital length of stay did not differ between receipt of amoxicillin/clavulanate or standard of care (10.3 days vs. 19.7 days, *p* = 0.237). Of patients whose cultures were collected as outpatients, 61% of amoxicillin/clavulanate recipients were admitted for therapy, whereas 89% of patients who received standard of care were admitted. Hospital length of stay for those who required admission from the community was also numerically shorter with amoxicillin/clavulanate (2.9 days) as compared to standard of care (8.4 days) (Table 2). 

Predefined subgroup analysis yielded no differences in clinical failure overall. None of the infection types (uncomplicated cystitis, complicated cystitis, and pyelonephritis) demonstrated a difference in outcomes between the two patient groups. Notably, in the subgroup of pyelonephritis patients, the clinical failure rates were numerically higher in the standard-of-care group (6/14. 42.9%) compared to amoxicillin/clavulanate: (6/14 [49%] vs. 0/5 [0%], *p* = 0.077) although the proportion of amoxicillin/clavulanate recipients treated for pyelonephritis was much lower than standard-of-care recipients (19.2% vs. 42.2%). There was a positive relationship between number of positive urine cultures in the past year and rates of clinical failure, but this did not differ between the two treatment arms. Immunocompromised patients, although more commonly receiving standard of care, had numerically identical rates of clinical failure in each arm (33.3%).

## 4. Discussion

In our study of adult patients with symptomatic UTI due to selected ceftriaxone non-susceptible Enterobacterales, we observed similar clinical failure rates with amoxicillin/clavulanate compared to other standard-of-care antibiotics. Additionally, patients who received amoxicillin/clavulanate for inpatient treatment of UTI had a numerically shorter duration of stay (2.9 + 1.2 days vs. 8.4 + 8.9 days, *p* = 0.053). Inpatients who received amoxicillin/clavulanate also had a shorter total duration of therapy than their counterparts receiving standard of care (10.3 + 9.8 days vs. 19.7 + 20.5 days, *p* = 0.237). Our data open the possibility of using a safe and well-tolerated oral antimicrobial for patients who may previously have had no oral options due to resistance or were not candidates for available oral agents. 

The role of oral beta-lactams in the treatment of urinary tract infections has become a contentious topic of interest among the infectious diseases community. Available evidence supports oral beta-lactams as an option for step-down therapy for UTIs, including cystitis and pyelonephritis. Sutton et al. conducted a large retrospective study of over four thousand patients from Veterans Affairs hospitals with bloodstream infections from a urinary source [15]. The study compared 30-day all-cause mortality and 30-day recurrent bacteremia among patients that were stepped down to an oral beta-lactam (OBL), fluoroquinolones (FQ), or trimethoprim-sulfamethoxazole (TMP-SMX). Propensity-based weights were used to adjust for population differences between groups. Of the 955 patients that received oral beta-lactams, 251 (26%) received amoxicillin/clavulanate. Mortality (OBL 3% vs. FQ/TMP-SMX 2.6%) and recurrent bacteremia (OBL 1.5% vs. FQ/TMP-SMX 0.4%) did not differ between the groups. For patients receiving amoxicillin/clavulanate, mortality and recurrent bacteremia were 5.2% and 1.6%, respectively. Saad et al. conducted a multicenter retrospective study of 207 patients from Canada [16]. They compared outcomes between patients who received either oral beta-lactams or oral ciprofloxacin after receipt of empiric IV beta-lactams for treatment of *E. coli* bloodstream infections caused by urinary tract infections. Propensity scoring was carried out to account for comorbid conditions, age, sex, and recurrent UTI risk factors. Patients received 5 days of IV beta-lactam and 7 days of oral antibiotics. Amoxicillin/clavulanate was used by 19 (25%) patients in the oral beta-lactam group. Clinical cure was 94% (72/77) in the oral beta-lactam group and 98% (127/130) in the ciprofloxacin group. Secondary outcomes of 30-day mortality, *Clostridioides difficile* infection, and length of hospital stay did not differ between groups. Mack et al. conducted a multicenter retrospective study of patients with Gram-negative bloodstream infections from a urinary source [17]. The study design was a noninferiority design powered to detect a 10% difference between groups. Patients received at least 48 h of IV antibiotic and one dose of either an oral beta-lactam or fluoroquinolone or TMP/SMX. Amoxicillin/clavulanate was used by 20 (17%) patients in the oral beta-lactam group. The primary outcome of 30-day all-cause hospital readmission occurred in 17/119 (14%) patients in the oral beta-lactam group and 15/91 (17%) patients in the fluoroquinolone and TMP/SMX group. Readmission with recurrent bloodstream infection occurred in four (3%) patients in the oral beta-lactam group and none in the fluoroquinolone and TMP/SMX group. Readmission with recurrent urinary tract infection occurred in six (5%) patients in the oral beta-lactam group and none in the fluoroquinolone and TMP/SMX group.

Many of these comparative studies, however, either excluded patients with resistant organisms or did not mention susceptibility patterns of the isolates, and there is a paucity of data that do evaluate the role of beta-lactams in resistant Enterobacterales urinary tract infections. The two largest studies to date retrospectively analyzed a retrospective study of non-carbapenem parenteral beta-lactams versus carbapenems for ESBL Enterobacterales urinary tract infections. Both analyses found similar clinical outcomes between groups, including in pyelonephritis patients [18,19]. One study also identified a trend toward increased incidence of carbapenem-resistant organisms in those who received carbapenem therapy [19]. In contrast, at least one oral carbapenem recently in development failed to achieve non-inferiority to fluoroquinolones. While sulopenem resulted in a similar clinical cure, discordance was observed whereby sulopenem-treated patients were unable to achieve microbiological eradication [20]. Subsequent analyses of such discordant events by the US FDA have suggested that failure of microbiological eradication is associated with increased risk of late clinical failures [14]. Taken together, these data highlight the need for continued study of oral beta-lactam treatment for UTIs. 

Pharmacokinetic (PK) and pharmacodynamic (PD) target attainment is of great importance, given the original hypothesis that amoxicillin/clavulanate’s utility is based on its ability to concentrate sufficiently at the site of infection due to its significant renal elimination. There is one PK/PD analysis of oral agents for pyelonephritis that, at face value, may spark concerns about amoxicillin/clavulanate use. In this analysis by Cattrall and colleagues, amoxicillin/clavulanate failed to achieve 90% cumulative fraction response (CFR) at the dosing regimen of 500 mg every 8 h. There are several key differences between this population and ours. Most notable is the MIC distribution of amoxicillin/clavulanate against *E. coli*. In the aforementioned study, authors incorporated the entire MIC distribution of isolates into their analysis, rather than excluding non-susceptible isolates. In this cohort, the MIC50 was 8/4 mg/L, with an MIC90 of 64/32 mg/L, and 38% of isolates were resistant using the EUCAST susceptibility breakpoint of <8 mg/L. Conversely, our institutional antibiogram demonstrates that 88% of all *E. coli* isolates are susceptible to amoxicillin/clavulanate at a breakpoint of <8/4 mg/L. Additionally, this analysis did not take urinary penetration into account, as the authors stated that pyelonephritis is an infection of the renal tissue rather than the collection system itself; therefore, pharmacokinetic and pharmacodynamic parameters may be more favorable when incorporating the high urinary penetration [21]. Indeed, a number of studies have evaluated the urinary pharmacokinetics of amoxicillin and found that typical oral doses result in high, prolonged urinary concentrations far in excess of the MICs of even highly amoxicillin-resistant bacteria [22]. Average urinary concentrations of 1100 mg/L were reported over the period of 0 to 6 h following a single 500 mg oral dose of amoxicillin in healthy subjects with normal renal function, while single 1000 mg oral doses produced a median of 3313 mg/L over 0–6 h after the dose. Such high concentrations are also achieved in patients with renal impairment; average urinary concentrations of 142 to 215 mg/L over 0–8 h were achieved in subjects with glomerular filtration rates as low as 10 mL/minute after administration of single doses of amoxicillin 500–1000 mg [22]. Oral administration of clavulanate likewise produces high urinary concentrations which (in combination with amoxicillin) are sufficient to result in bactericidal activity over periods of four hours or more, even against amoxicillin-non-susceptible strains of *E. coli* [23]. The high urinary concentrations of amoxicillin and clavulanate achieved after oral dosing thus indicate excellent PK/PD characteristics for the treatment of UTI and highlight the need for urine-specific breakpoints for this agent. In evaluations of other commonly used parenteral antibiotics, high rates of target attainment have been observed in the urine, even among isolates deemed resistant to the organism using systemic breakpoints [23]. In consideration of these principles, there is a need for high-quality data to more firmly establish the correlation of amoxicillin/clavulanate MIC, urinary PK-PD target attainment, and in vivo microbiological eradication. 

The potential use of an oral antibiotic to treat urinary tract infections caused by organisms with in vitro resistance to that agent is not unique to amoxicillin/clavulanate for ESBL-producing pathogens. Favorable clinical outcomes have also been previously reported with the use of aminopenicillins in the treatment of UTIs caused by ampicillin-resistant vancomycin-resistant enterococci (VRE). A study by Cole and colleagues reported clinical cure rates of 86% (12/14) with aminopenicillins vs. 70% (16/23) with non-beta-lactams (linezolid, daptomycin, or fosfomycin) among hospitalized patients infected with VRE with ampicillin MICs ranging from 128 to 512 mg/L. The majority of these patients were treated with oral amoxicillin [24]. A similar study by Shah and colleagues reported clinical cure in 88% (74/84) and microbiological eradication in 86% (50/58) of evaluable hospitalized patients with UTI caused by ampicillin-resistant VRE [25]. Although the majority of the patients in this study were treated with intravenous ampicillin, patients treated with oral ampicillin also had favorable clinical and microbiological outcomes. A third study by Richey and colleagues reported clinical cure in all eight patients (100%) with UTI caused by ampicillin-resistant *Enterococcus faecium* and treated with oral amoxicillin; clinical cure was 78% (7/9) among patients receiving nitrofurantoin [26]. All of these studies were limited by small sample sizes; however, the consistently favorable results do suggest that aminopenicillins are potentially reasonable treatment options for UTI in patients infected with ampicillin-resistant enterococci, including those with comorbidities such as diabetes and chronic kidney disease. The results of these studies are again likely explained by the exceedingly high, prolonged urinary concentrations of these drugs and lend support to the feasibility of the use of amoxicillin/clavulanate for treatment of UTI caused by ESBL-producing uropathogens. 

Our data open the possibility for oral beta-lactam therapy in an ESBL infection where traditionally patients have been confined to either parenteral therapy or non-beta-lactam oral therapeutic classes that subject the patient to collateral damage. We anticipated to see more benefits in length of stay by the incorporation of an alternative oral option; however, US-based antimicrobial surveillance data show that only 44% of ceftriaxone non-susceptible *Escherichia coli* remain susceptible to amoxicillin/clavulanate, highlighting the importance of avoiding empiric use of amoxicillin/clavulanate if the clinical suspicion of a resistant organism is high [27]. Therefore, admission still may be required until sensitivities are confirmed for patients with a history of resistance, high risk of resistance, or for patients with complicated cystitis with systemic symptoms of infection or pyelonephritis. The benefit of oral beta-lactams in preventing adverse events associated with alternative therapeutic classes is still well-described in the literature but was not assessed in this analysis.

Our study is unique in that it is the first analysis of amoxicillin/clavulanate for ceftriaxone non-susceptible UTIs that included a comparator group. A previous analysis by Beytur and colleagues included 46 patients who received amoxicillin/clavulanate for UTIs caused by an ESBL-producing organism; however, no comparator group was evaluated. Investigators also included patients with other infections, such as prostatitis (n = 10, 22%). Another key limitation of Beytur and colleagues’ study is the unclear timeline of follow-up. In our cohort, the mean time to clinical failure was over 30 days; thus, it is unclear if patients in the previous analysis were followed long enough to identify clinical failures [8]. Additionally, our analysis included a broad range of patients, including those with recurrent UTIs, uncomplicated pyelonephritis, and immunocompromised hosts including renal transplant recipients and thoroughly assessed lead-in therapy, which was not addressed in the prior analysis.

Another unique consideration surrounding these findings is the sample size itself. During the time period queried—1 January 2012 to 1 August 2022—there were no guideline recommendations to support amoxicillin/clavulanate for ceftriaxone non-susceptible isolates. In fact, the paper published by Beytur et al. was not published until early 2015. Ostensibly, providers were either not identifying these isolates as phenotypically ESBL, unaware that there is no evidence for amoxicillin/clavulanate in such isolates, or both. Of the nineteen amoxicillin/clavulanate recipients who were admitted at some point in their treatment course, nine of them (47%) had at least one day of amoxicillin/clavulanate while in the hospital as opposed to a new medication on day of discharge. Notably, our microbiology lab does not demarcate ceftriaxone-resistant isolates as potential ESBL isolates, nor are the susceptibility results presented in a cascaded or selective manner. Although the IDSA/SHEA guidelines for antimicrobial stewardship recommend implementing cascade and selective reporting of antibiotics, it is listed as a weak recommendation supported by low-quality evidence [28]. In one French prospective randomized controlled case vignette study, appropriate prescribing of antimicrobials for urinary tract infections was significantly higher when cascade and selective reporting was employed [29]. Although we do not have a comparative time period where this was implemented, one may be curious about the impact of prescribing.

This study is not without limitation. The sample size is small and is underpowered to detect a difference in the primary outcome of clinical failure. Additionally, the retrospective nature introduces selection bias to the dataset, although this was not identified in baseline characteristics with the exception of more immunocompromised hosts in the standard-of-care group. Additionally, a retrospective chart review introduces the possibility that diagnoses were misclassified or that information was inappropriately documented in the chart. In regard to treatment, lead-in therapy was allowed so long as it did not exceed half of the therapy duration, which may have masked the true effect of definitive therapy. Although no data exist that single-dose carbapenems are adequate to cure either cystitis or pyelonephritis, such data do exist for other agents such as ceftriaxone [30,31], fosfomycin [32] and aminoglycosides [33]. Therefore, the true impact of single- or multiple-dose active lead-in therapy is not known. Another limitation of this analysis is the lack of confirmation of ESBL isolates. We relied on ceftriaxone non-susceptibility as a surrogate which may be overly sensitive for the detection of ESBL, particularly CTX-M class, enzymes. Following the Clinical and Laboratory Standards Institute (CLSI) recommendations, our laboratory does not use genotypic confirmation, nor phenotypic confirmation with CLSI’s suggested ceftazidime plus clavulanate disc diffusion. Therefore, our population of ceftriaxone non-susceptible isolates may be overly sensitive for ESBL identification, yet this does represent real-world microbiology laboratory procedures. Finally, the retrospective nature of this analysis made routine follow-up urine cultures to assess for microbiologic cure implausible, although follow-up urine cultures for test-of-cure are generally not recommended in routine clinical practice.

## 5. Conclusions

Our results suggest that amoxicillin/clavulanate may be an appropriate alternative therapy for urinary tract infections caused by ceftriaxone non-susceptible Enterobacterales, and that recipients may require fewer days of hospitalization. No differences in outcomes were identified in notable subgroups, including category of urinary tract infection, immunocompromised hosts, and amoxicillin/clavulanate MIC. This study supports the growing body of evidence to support site-specific PK/PD values in selecting antimicrobial therapy and ideally will lay a foundation for future evaluations. Although results are promising, larger studies with improved ability to control for confounders should be conducted for confirmation. 

## Figures and Tables

**Figure 1 pharmacy-12-00060-f001:**
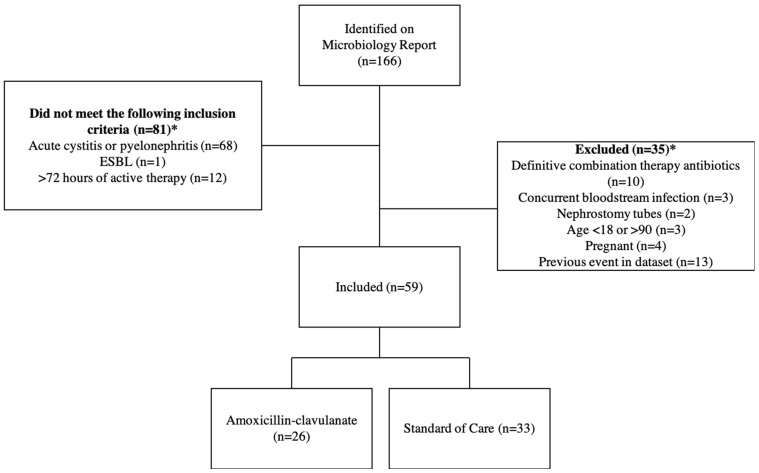
Cohort selection. * Patients may have met more than one criterion.

**Table 1 pharmacy-12-00060-t001:** Baseline characteristics.

	Amoxicillin/ Clavulanate (n = 26)	Standard of Care (n = 33)	Total (n = 59)	* p *-Value
**Age (years)**, (mean (SD))	62.2 (18.0)	58.2 (17.5)	60.0 (17.7)	0.396
**Male** (n (%))	11 (42.3)	9 (27.3)	20 (33.9)	0.226
**Race** (n (%)) White Black/African American American Indian/Alaskan Asian Hispanic	11 (42.3) 7 (26.9) 1 (3.8) 2 (7.7)5 (19.2)	17 (51.5) 2 (6.1) 0 (0.0) 0 (0.0) 14 (42.4)	28 (47.5) 9 (15.3) 1 (1.7) 2 (3.4)19 (32.2)	**0.014**
**Infection Type** (n (%)) Uncomplicated cystitis Complicated cystitis Uncomplicated pyelonephritis	3 (11.5) 18 (69.2) 5 (19.2)	4 (12.1) 15 (45.5) 14 (42.2)	7 (11.9) 33 (55.9) 19 (32.2)	0.142
Inpatient at time of urine culture (n (%))	18 (69.2)	18 (54.5)	36 (61.0)	0.251
**Urinary tract abnormalities** (n (%))	12 (46.2)	17 (51.5)	29 (49.2)	0.683
**Positive urine cultures within previous 12 months** (n (%)) **0** **1** **2** **3+**	13 (50.0) 9 (34.6) 1 (3.8) 3 (11.5)	8 (24.2) 13 (39.4) 3 (9.1) 9 (27.3)	21 (35.6) 22 (37.3) 4 (6.8) 12 (20.3)	0.160
**Charlson Comorbidity Index** (mean (SD))	4.5 (3.3)	3.7 (1.9)	4.1 (2.6)	0.293
**Immunocompromised** (n (%)) Renal Transplant	3 (11.5) 1 (3.8)	12 (36.4) 6 (18.2)	15 (25.4) 7 (11.9)	**0.030** 0.091
**Serum WBC** (cells/uL) (mean (SD))	10.3 (5.8)	9.7 (5.2)	10.0 (5.4)	0.673
**Serum lactate** (mmol/L) (mean (SD))	2.0 (0.9)	2.6 (1.3)	2.4 (1.2)	0.175
**Pyuria** (n (%))	24 * (100.0)	31 (93.9)	55 (96.5)	0.220
**Charlson Comorbidity Index** (mean (SD))	4.5 (3.3)	3.7 (1.9)	4.1 (2.6)	0.293
**Treatment duration** (days) (mean (SD))	7.0 (2.7)	9.3 (5.2)	8.3 (5.0)	0.085
**Active lead-in therapy** ** (n (%))	6 (31.6)	13 (65.0)	19 (48.7)	**0.037**
**Organism** (n (%))				
* E. coli *	23 (88.5)	25 (75.8)	48 (81.4)	0.214
* K. pneumoniae *	3 (11.5)	8 (24.2)	11 (18.6)	0.362

SD: standard deviation, WBC: white blood cells. * Two patients missing data. ** Not all patients received lead-in therapy.

**Table 2 pharmacy-12-00060-t002:** Primary and secondary outcomes.

	Amoxicillin/Clavulanate (n = 26)	Standard of Care (n = 33)	*p*-Value
**Clinical failure** (n (%)) New symptoms Retreatment Both	5 (19.2) 0 (0.0) 3 (11.5) 2 (7.7)	10 (30.3) 3 (9.1) 6 (18.2) 1 (3.0)	0.332
**Time to failure (days)** (Mean (SD))	31.2 (23.8)	35.0 (24.6)	0.725
**Hospital length of stay (days)**, (Mean (SD))			
Admitted from Community	2.9 (1.2)	8.4 (8.9)	0.053
Inpatient at Diagnosis	10.3 (9.8)	19.7 (20.5)	0.237
**Amoxicillin/Clavulanate-resistant organism** (n (%))	3 (11.5)	2 (6.1)	0.453
**Carbapenem-resistant organism** (n (%))	1 (3.8)	0 (0.0)	0.441

## Data Availability

The data presented in this study are available in the article.

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
