# Peer review of "Evaluation of Oral Amoxicillin/Clavulanate for Urinary Tract Infections Caused by Ceftriaxone Non-Susceptible Enterobacterales"

_pharmacy, 2024, doi:10.3390/pharmacy12020060_

Round 1
Reviewer 1 Report
Comments and Suggestions for Authors
Dear Ms. Anne. Zhang
Assistant Editor, MDPI
I am sending my review comments to the manuscript Pharmacy Number- 2784739 entitled: Evaluation of oral Amoxicillin/Clavulanate for Urinary Tract Infections Caused by Cefriaxone non-susceptible Enterobacteriales
Comments to the Author
The manuscript prepared by Madison E. Salam and co- workers highlight the need for continued study of oral beta lactam treatment for UTIs.
‘This study is not without limitation. The sample size is small and is underpowered to detect a difference in the primary outcome of clinical failure.’
In my opinion the method of presenting the research and its results is not clear.
Text in the papaer is structured in the way that it is difficult read. Sentences are too long, punctation marks are missing
Conclusion is too short.
After careful reading of this paper in my opinion the work is suitable for Pharmacy after making corrections to the form of presenting research results.
With Regards
Reviewer 2 Report
Comments and Suggestions for Authors
Generally speaking, the manuscript is well written and the study is well-designed. I have no other comments and recommed the publication in present form.
Reviewer 3 Report
Comments and Suggestions for Authors
Title: Evaluation of Oral Amoxicillin/Clavulanate for Urinary Tract Infections Caused by Ceftriaxone Non-Susceptible Enterobacterales
By Ceftriaxone non-susceptible, do you mean ceftriaxone resistant? If so, I recommend stating it conventionally for keyword search
Line 19 of the abstract, I suggest “Gram-negative microorganisms” instead of organisms
There is a lack of clarity for non-specialist. Ex: “Ambler Beta-lactamases class A enzyme” sounds better than “Ambler class A enzyme”
Line 20: “therapies” not “thera-pies”
I don’t understand why there are so many hyphens in the text even on the same line.
Overall, the abstract should be improved, the results should be clearly described despite the shortness.
Introduction
I am not aware of the citation style \cite(1,2) and some time the “.” is before/after the citation, it is confusing.
Line 54: what is the meaning of “...patient are exposed to significant contact with ..” please rephrase the sentence.
Line 65: do you mean "..there is a need …? More option or alternative option??
What is the meaning of Standard of care in this context? What is the guideline for the ESBL treatment? Can the author state it in the text?
MM
All the targeted bacteria species are Enterobacteriaceae, why do the authors have to include up to order?
Line 82: define IRB the first time it is mentioned in the text
Line 86: how do the authors differentiate between AmpC related resistance and ESBL? Were the genome sequenced? PCR screening?
Line 86-87: if Serratia and Citrobacter are less prevalent, why do you have to exclude theme as criteria?
The authors should explain why they have to exclude some Carbapenemase-producing causal bacteria patients.
Line 92-93: of course Uncomplicated is the opposite of complicated
Line 121 -124: what about patients that develop pus with ESBL with different microorganisms within 90 days?? The statement should be scientifically concise.
Results
What is the lexical difference between “Did not meet the following inclusion criteria” and “Excluded”?
Line 142: provide sex ratio.
Line 142: 64.4% of patients had at least one positive urine culture in the preceding year?? After excluding patient with 90 days treatment? Why do you have to include confusing statement?
Table:
Age format: Age(years), (mean, SD): the figures should be : 62.2, 18.0
Where are the majority of patients? Female! in the table?
Race? Was it part of the criteria? Significant?
Overall the main outcome is interesting, reducing the patient duration of stay and therapy with amoxicillin + clavulanic acid
please improve the English, some statements are scientifically confusing, rephrase them, remove ambiguous sentences and correct typo.
